# Monitoring Illegal Logging Using Google Earth Engine in Sulawesi Selatan Tropical Forest, Indonesia

A. Mujetahid [1,*], Munajat Nursaputra [2] and Andang Suryana Soma [3]

1   Forest Harvesting Laboratory, Faculty of Forestry, Hasanuddin University, Makassar 90245, Indonesia
2   Forestry Planning and Information System Laboratory, Faculty of Forestry, Hasanuddin University, Makassar 90245, Indonesia; munajatnursaputra@unhas.ac.id
3   Watershed Management Laboratory, Faculty of Forestry, Hasanuddin University, Makassar 90245, Indonesia; s_andangs@unhas.ac.id
*   Correspondence: mujetahid@unhas.ac.id

**Abstract:** Forest destruction has been found to be the cause of natural disasters in the form of floods, landslides in the rainy season, droughts in the dry season, climate change, and global warming. The high rate of forest destruction is caused by various factors, including weak law enforcement efforts against forestry crimes, such as illegal logging events. However, in Indonesia, illegal logging is only discovered when the perpetrator has distributed the wood products. The lack of monitoring of the overall condition of the forest has an impact on the current high level of forest destruction. Through this research, the problems related to environmental damage due to illegal logging will be described through remote sensing technology, which is currently mainly developed on the basis of artificial intelligence and machine learning, namely Google Earth Engine (GEE). Monitoring of illegal logging events will be analysed using Sentinel 1 and 2 data. Obtaining satellite imagery with relatively small cloud cover for tropical regions, such as Indonesia, is remarkably difficult. This difficulty is due to the presence of a radar sensor on Sentinel 1 images that can penetrate clouds, allowing for observation of the forest condition even in the presence of clouds. Using the random forest classification algorithm of the GEE platform, data on forest conditions in 2021 were obtained, covering an area of 2,843,938.87 ha or 63% of the total area of Sulawesi Selatan Province. An analysis using a map of the function of forest areas revealed that of the current forest area, 38.46% was non-forest estates and 61.54% was forest areas. The continued identification of illegal logging events also found 1971 spots of forest change events in the vulnerable time of the first period (January–April) with the second period (April–August), and 1680 spots of forest change events in the vulnerable period of the second period (April–August) with the third period (September–December), revealing a total incident area of 7599.28 ha.

**Keywords:** illegal logging; machine learning; GEE; Sentinel; Sulawesi Selatan

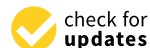



## 1. Introduction

In the last decade, the issue of Indonesia's forest destruction has attracted sufficient public attention. Various groups have raised awareness and enthusiasm regarding the importance of organising and restoring forest quality. Forest destruction has been punished by the community as it is a cause of natural disasters in the form of floods and landslides in the rainy season and drought in the dry season, as well as being responsible for climate change and global warming. The high rate of forest destruction is caused by various factors, including mismanagement, illegal logging, forest and land fires, forest land conflicts, mining, encroachment, conversion of forest areas for other uses that do not meet the applicable rules, and weak law enforcement efforts against forestry crimes [1–3].

If this trend persists, the highly diverse tropical forests of Indonesia are predicted to disappear in the next few years [4]. Therefore, the whole world is highly focussed on

making the issue of forest degradation in Indonesia an important part of negotiations in order to develop bilateral or multilateral cooperation. Activities that have contributed to the increase in the deforestation rate are cultivation, agricultural/food development and plantations (22%), infrastructure development (16%), illegal activities in forest areas due to open access (shrubs) (61%), settlements and transmigration (0.4%), and mining (0.6%). Illegal activities in forest areas due to open access have the highest contribution among the others because strengthening the forest areas of Indonesia has not been achieved. Thus, illegal logging will have economic and ecological impacts [5]. Regarding the economic sector, the state's loss has caused a reduction in state income on taxes from the value of the wood. On a broad scale, there will be a loss of opportunities to take advantage of the diversity of products in the future.

The substantial losses that cannot be assessed are those due to the disruption of environmental functions caused by the loss of forest stands, which will result in environmental damage, climate change, decreased land productivity, erosion and flooding, habitat destruction, and loss of biodiversity. However, illegal logging is only discovered in Indonesia when the perpetrators have distributed their timbre products. The current high amount of forest destruction is attributed to the lack of monitoring of the overall forest condition. This phenomenon is based on the limited human resources and insufficient budget for routine patrols.

Through this research, the problems related to environmental damage due to illegal logging will be described via remote sensing technology, which is currently mainly developed on the basis of artificial intelligence and machine learning [6,7]. Many platforms that currently use artificial intelligence and machine learning can be developed to monitor forest conditions regularly. One of the most powerful platforms is Google Earth Engine (GEE) [8,9]. This platform was accidentally created by a developer at Google who was concerned about the destruction of the forest around her residence. With the help of the remote sensing image database of NASA, she attempted to display imagery differences over time and was finally able to provide evidence to the court with accurate data. This technology was initially only considered to be a toy by some people. However, the data have raised awareness in the community and the government regarding forest destruction.

GEE is currently a computing platform for remote sensing and geographic information processing, and is extensively used worldwide [10–12]. This platform not only provides basic computational functions for raster and vector data, but also introduces application programming interfaces based on JavaScript and Python programming languages to allow users to conduct high-efficiency studies over a large area easily [13]. The availability of Sentinel imagery data, both Sentinel 1 and 2, in the GEE database is very helpful for monitoring the condition of forests in Indonesia, which is a tropical region. Obtaining satellite images with relatively little cloud cover for a tropical region such as Indonesia is very difficult. So, with the radar sensor presented in Sentinel 1 imagery that can penetrate the clouds, it allows us to see the condition of the forest, even when there are clouds. Sentinel Image Data which is also produced in series is very helpful in regular monitoring. The existence of a combination of data utilization and data processing technology in GEE can be a tool for the wider community to report damage occurring in their environment. The author of the current study aims to develop GEE by creating a wild logging detection algorithm that utilises multi-time remote sensing image data. Therefore, early monitoring of illegal logging can detect the locations of incidents. These data and information are expected to facilitate easy monitoring at the field level for future policymakers by providing information to increase patrols in areas with illegal logging potential.

## 2. Materials and Methods

### 2.1. Research Location

The research site is administratively located in Sulawesi Selatan Province, which is one of the provinces on Sulawesi Island that is responsible for maintaining 40% of the forest area on Sulawesi Island [14]. Sulawesi Selatan Province has a forest area of 50% of the total

area of Sulawesi Selatan Province, of which 22% is protected forest area, 12% is production forest area, and 16% is conservation forest area. Historically, the recorded events have often occurred in findings of illegal logging occasions within the forest area, as indicated by the distribution presented on the research location map in Figure 1.

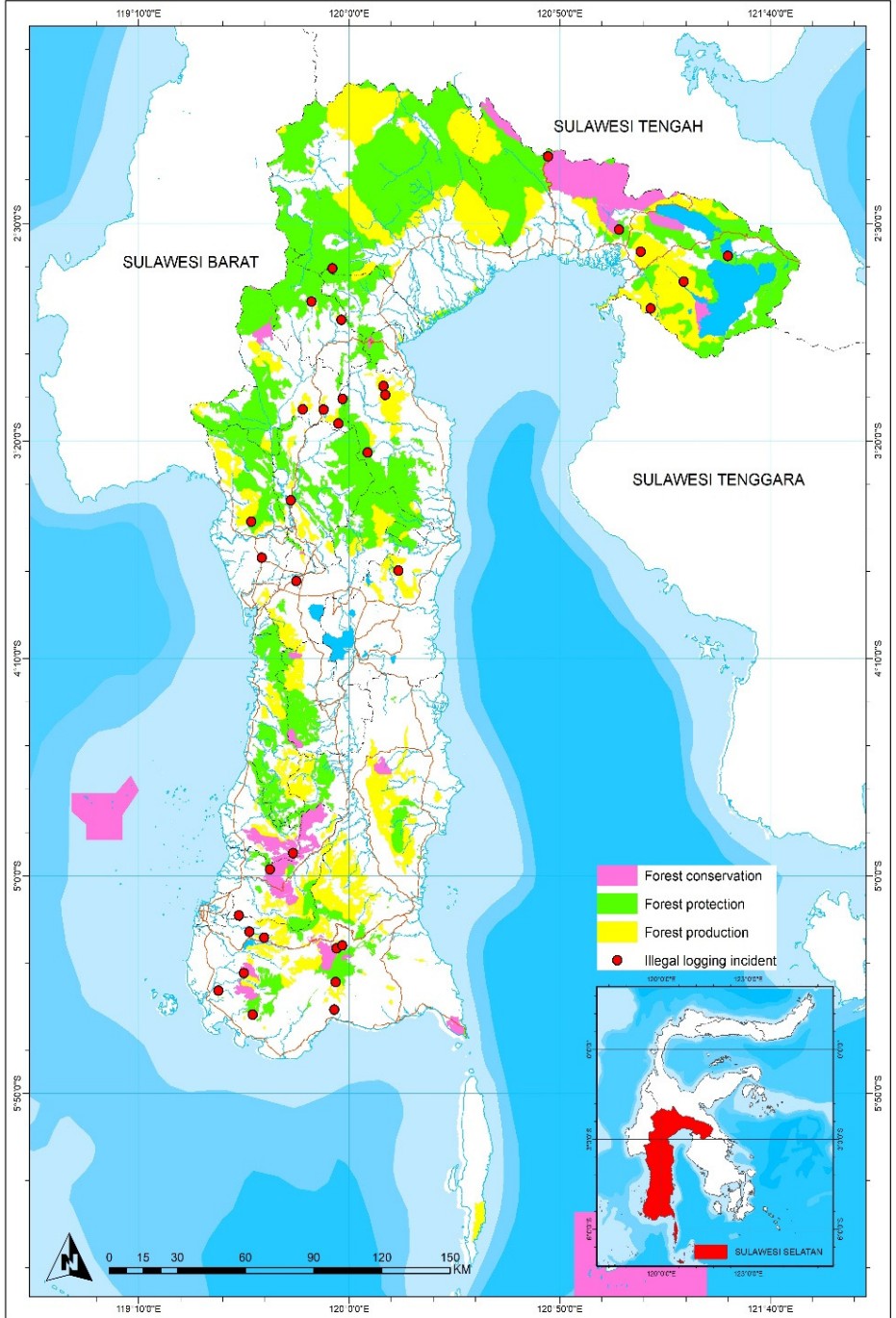

**Figure 1.** Research Llocation.

### 2.2. Data Collection

One of the main achievements of this study was in the utilisation of remote sensing data to monitor forest conditions in an area. However, one of the limitations of remote sensing data in Indonesia, which is the tropics, lies in the presence of clouds. Obtaining satellite imagery with a relatively low cloud cover, below 10%, in Indonesia is challenging.

The SAR sensors on Sentinel images using radar waves, which penetrate the clouds, allow for the observation of the Earth's surface, even in the presence of clouds [15,16].

Therefore, through this research, illegal logging event monitoring will be analysed using data from Sentinel 1 SAR and Sentinel 2 Multispectral. Sentinel 2 imagery was used to map forested areas, then in this forested area, using Sentinel 1, an analysis of the changes from several time periods was carried out to map areas where illegal logging occurred. The data used in this article used Sentinel 1 and 2 image data from 2021. Sentinel-2 was the product used in Sentinel-2 surface reflectance (https://developers.google.com/earth-engine/datasets/catalog/COPERNICUS_S2_SR (accessed on 20 April 2022)) and Sentinel-1 was the GRD product used in IW mode on polarisation VV than VH (https://developers.google.com/earth-engine/datasets/catalog/COPERNICUS_S1_GRD (accessed on 20 April 2022)), as presented in Figure 2.

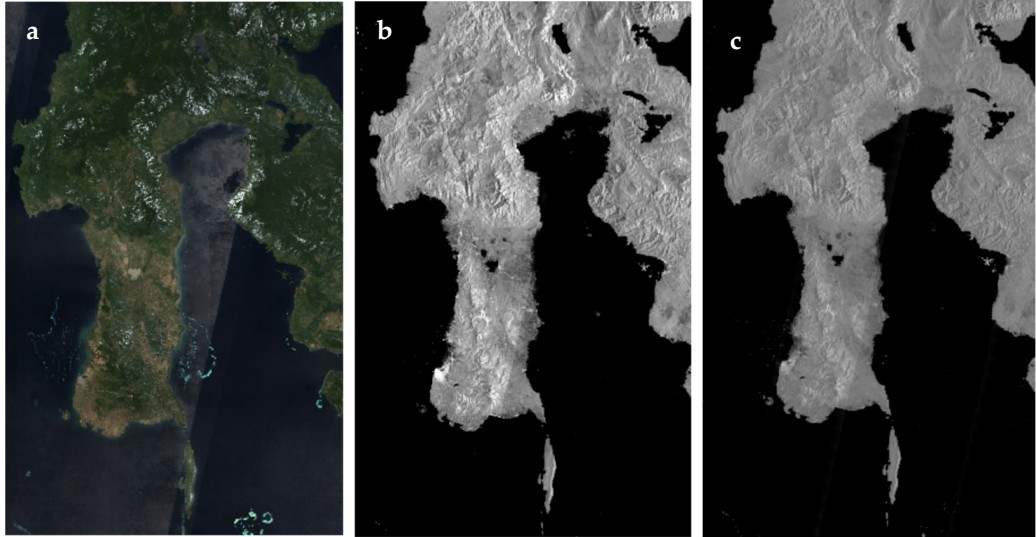

**Figure 2.** Sentinel 2 Multispectral True Colour (**a**), Sentinel 1 Polarisation VV (**b**) and VH (**c**) in Sulawesi Selatan Province.

### 2.3. Data Analysis

The research flow is divided into several stages, namely pre-processing, classification, and validation, as presented in the Figure 3. The entire process was conducted using the GEE platform through the code.earthengine.google.com page via the compiled algorithm. The GEE platform is a cloud computing-based machine learning platform where machine learning is one of the applications of artificial intelligence. Machine learning primarily aims to handle high-dimensional data, such as remote sensing data, and to map it into multiple classes with complex characteristics [17]. GEE provides numerous machine learning techniques, namely classification and regression tree (CART), random forests, and several other types of image classification [18].

### 2.3.1. Pre-Processing

A subset process was conducted to adjust the coverage of the analysed imagery according to the location of the study. This subset process aimed to accelerate the processing because the process would run fast in a small area. On the GEE platform, the subset process was performed by including the regional boundaries of Sulawesi Selatan Province as areas of interest in the subset algorithm, which was built using the following algorithm.

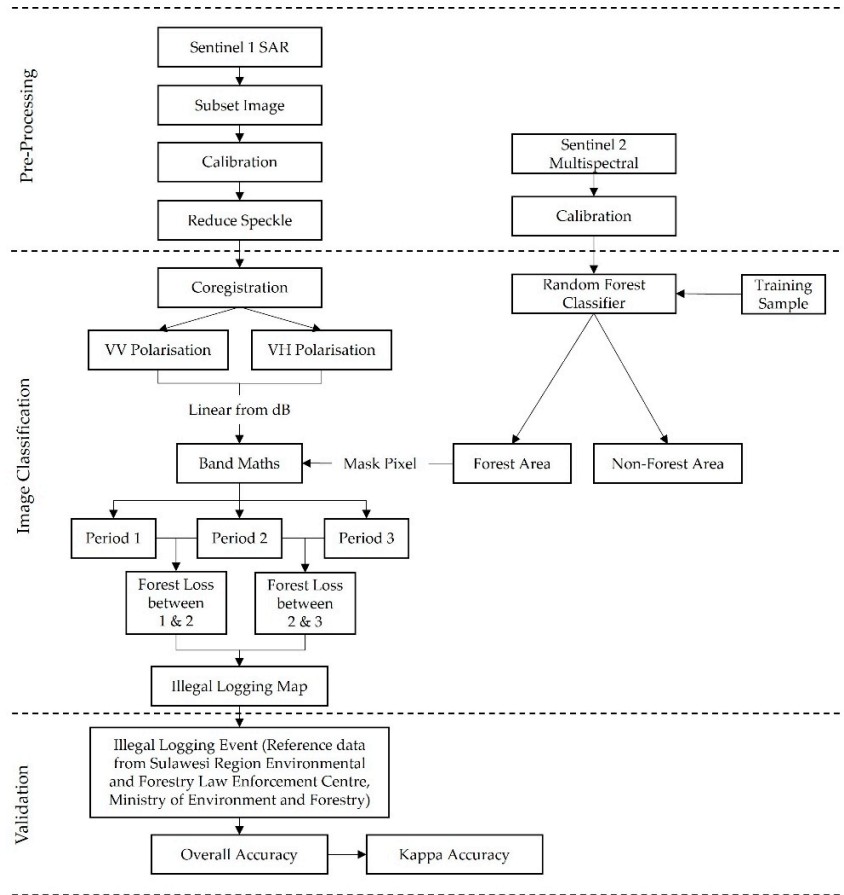

**Figure 3.** Research framework for illegal logging mapping.

Furthermore, the radiometric calibration process was conducted by processing the image components, thus providing a direct approach. The data used for forest cover classification for Sentinel Image two had less than 20% cloud cover. Several methods to minimise cloud cover on Sentinel 2 images were performed in GEE with philtre masking. Filter masking was used by combining satellite images contained in several periods of satellite imagery recording until an image with a clear and clean layer was obtained. The masking process was conducted by replacing cloud-covered image pixels with pixels on other images not covered by clouds by using the QA60 (Cloud Mask) band contained in Sentinel 2 satellite images.

Sentinel 1 data for Sentinel 1 images needed to be calibrated in order to obtain satisfactory data. Calibration is a procedure used for the radiometric correction of Sentinel-1 imagery by correcting pixel values representing radar backfilling of the reflected surface. A calibration vector was inserted as an annotation in the product, thereby allowing for simple conversion of image intensity values into sigma-naught, gamma-nought, or beta-naught values [15,19–21].

The Sentinel 1 data still had interference, namely speckle, or what is commonly considered noise, even after the calibration process. The disturbance was usually in the form of black and white spots caused by wave disturbances reflected in the large distribution of elements. Speckle filtering is a process used to improve image quality by removing speckles. Several speckle filtering methods, such as the Lee speckle philtre method, could be used in the current study to enhance the quality of Sentinel-1 imagery [22]. Lee is a method based on the minimum mean square error and geometric aspects. The Lee philtre is a statistical philtre designed to remove noise while still maintaining the quality of the pixel points and edge borders on the image.

2.3.2. Image Classification

1. Forest Cover Classification

The mapping of forested areas in this study used a supervised classification method with a random forest algorithm. Random forest was an improvement of the CART algorithm by applying bagging and random feature selection to machine learning decision tree algorithms, randomly selecting several features in each iteration [23]. The resulting tree had as many iterations as possible to resemble a forest. This study utilised random forests to handle high-dimensional data and multicollinearity due to its insensitivity to overfitting [24]. The classification of supervised random forest involved intensive interaction of the analysts, where the process of identifying objects in the image (training area) was conducted. The sampling of each sample needed to be performed considering the spectral pattern at each specific wavelength so as to obtain a good reference area to represent a particular object [25]. Supervised classification began by creating a training area based on the forest cover class and then continued with the classification process with a random forest algorithm. The land cover used was divided into the following two classes: forest and non-forest areas. The algorithm used for classifying random forests on the GEE platform was as follows.

*var classifier = ee. Classifier.smileRandomForest*

2. Classification of Illegal Logging Events

The classification method used to detect the occurrence of illegal logging in this study was a segmentation process for the average comparison of VV and VH polarisation images, which were divided into three data for the periods of January to April (period 1), May to August (period 2) and September to December (period 3) in areas detected as forested areas according to the first stage [26–29]. The image results were then calculated as the ratio to three images in order to obtain changes between periods 1 and 2 and those between periods 2 and 3. Forest spots that changed on the basis of the ratio calculation of the three image periods were detected as areas indicating the occurrence of illegal logging events. The algorithm for classifying illegal logging events on the GEE platform was as follows.

*var forestloss1= first_period.subtract(second_ period);*
*var forestloss2= second_ period.subtract(third_ period);*

The obtained image change conditions were overlaid with a map of forest distribution and the function of forest areas in Sulawesi Selatan Province to comprehensively explore the forest changes that occured either in forest areas or other areas.

2.3.3. Validation

Classified data require some validation to evaluate their accuracy and suitability for future use [30]. Conducting field surveys and validating outputs were generally impossible. The reference data were obtained from the Sulawesi Region Environmental and Forestry Law Enforcement Centre, Ministry of Environment and Forestry, to conduct accuracy tests using external data sources as the references. Illegal logging monitoring data obtained from relevant agencies during 2021 totalled 370 observation points. That point data were then tested with the results of the data processing of illegal logging incidents obtained at GEE, where testing was carried out at 500 points, namely 370 points where illegal logging occurred and 130 points that were distributed as training samples. The accuracy test was utilised to determine the extent of the accuracy of the image interpretation results. The accuracy test was a comparison between two data from the image classification results with the field conditions. The method for calculating the value of accuracy of the image interpretation results aims to test the magnitude of accuracy with kappa accuracy. The acceptable accuracy of the image classification was 85% [31]. According to Jaya, the currently recommended accuracy was kappa accuracy because the overall accuracy is still generally overestimated [32]. The algorithm used for the kappa accuracy test on the GEE platform is as follows.

```
var Accuracy = validation.errorMatrix('Landcover', 'classification');
print('Confussion matrix', Accuracy);
print('Overall accuracy', Accuracy.accuracy());
print('Consumer accuracy', Accuracy.consumersAccuracy());
print('Producer accuracy', Accuracy.producersAccuracy());
print('Kappa statistic', Accuracy.kappa());
```

## 3. Results and Discussion

### 3.1. Forest Cover Identification

The use of GEE in mapping forest conditions in Sulawesi Selatan Province was constructive for producing data on forest conditions at any time or annually [9,33]. Using the random forest classification algorithm, data on forest conditions in 2021 in Sulawesi Selatan Province were obtained, covering an area of 2,843,938.87 ha or 63% of the total area of Sulawesi Selatan Province, as presented in the distribution in Figure 4 [24,34,35]. Data on the existing condition of this forest could be a baseline for measuring the rate of forest change in the future, especially to observe the area of existing forests for each administrative area, as presented in Figure 5.

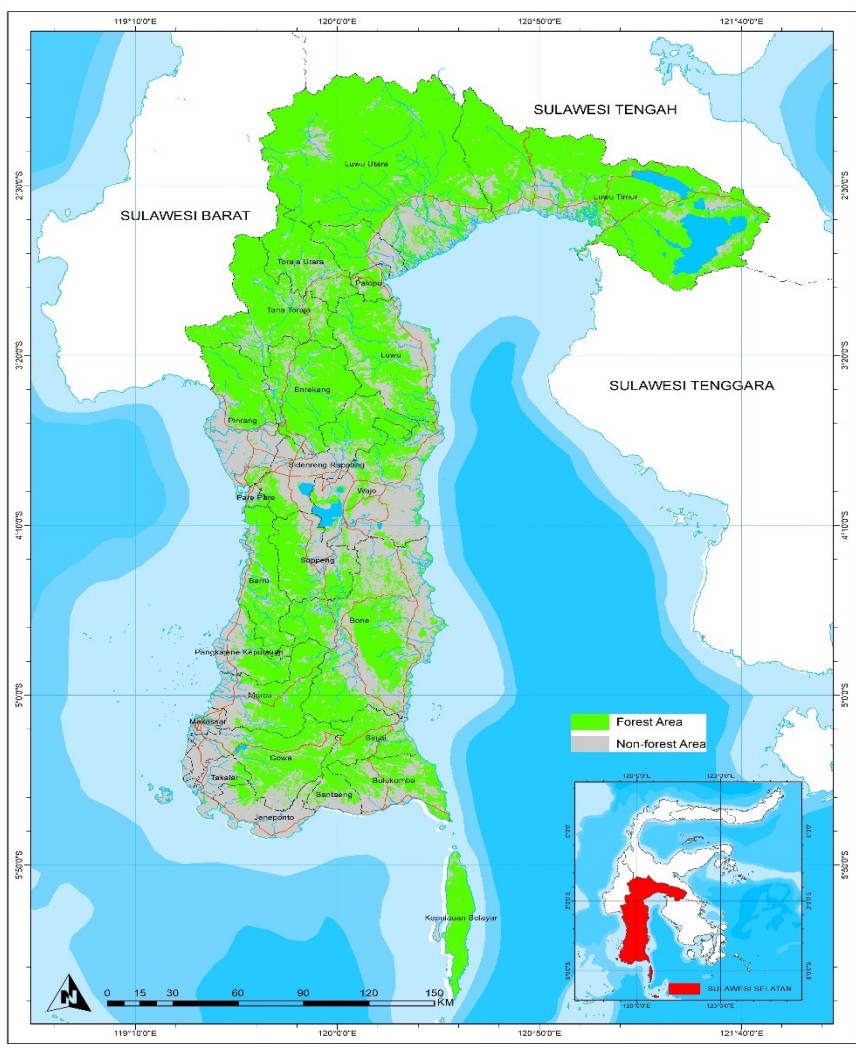

**Figure 4.** Map of forest distribution in Sulawesi Selatan Province.

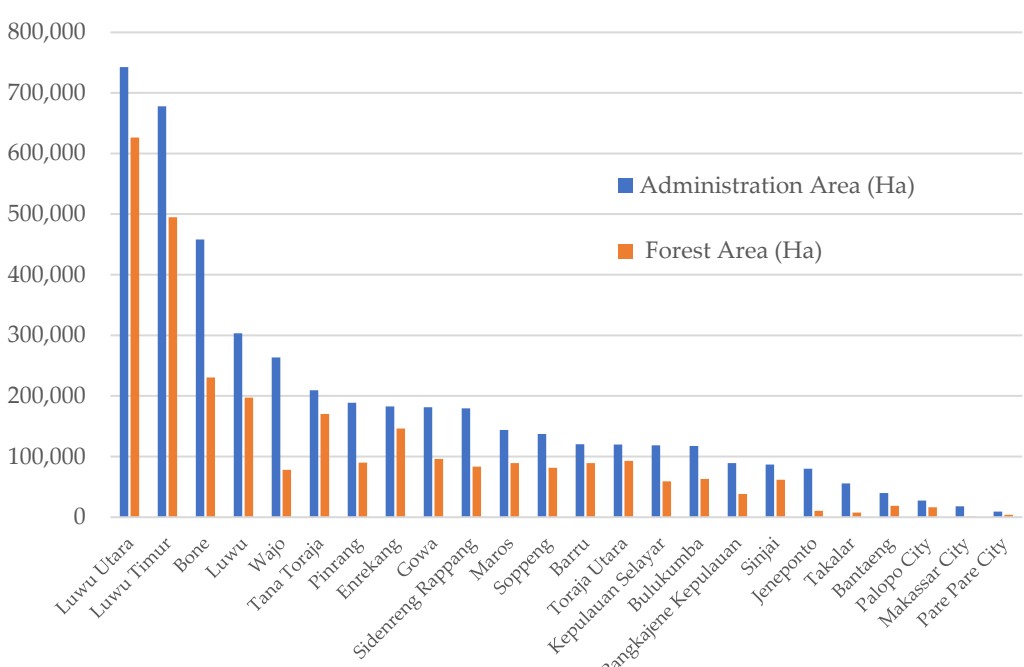

**Figure 5.** Distribution of forest area in each regency/city area.

Figures 4 and 5 show that the largest forest area was in North Luwu Regency, with an area of 84% of the total administrative area of North Luwu Regency, followed by East Luwu Regency with an area of 73% of the total administrative area. Of the 24 regencies/cities in Sulawesi Selatan, four regencies/cities, namely Wajo, Jeneponto, Takalar, and Makassar City, had a forest area less that 40% compared with that of their administrative areas. This condition caused the failure of several areas to meet the policy direction of realising the preservation of permanent vegetation by 40% in each area of Sulawesi Island according to the Spatial Plan for the Sulawesi Island area law. The forest area interpreted on the GEE platform was the overall area contained in the designated area by the Government of Indonesia as a function of forest and non-forest areas [36–38]. Based on the analysis results of the forest area function map of Sulawesi Selatan Province, 38.46% (around 1,093,841.79 ha) of the current forest area was in an area designated as a non-forest estate, which was almost spread throughout the regency/city area, especially for Makassar City, whose entire forest was in a non-forest estate area. Meanwhile, 61.54% was in forest areas with 35.42% in protected forest areas, 20.24% in production forest areas, and 5.88% in conservation areas located in Palopo City, Tana Toraja Regency, Soppeng, Sinjai, Sidendreng Rappang, Pangkajene Islands, Maros, East Luwu, Jeneponto, Gowa, Bulukumba, Bone, and Barru, as presented in Figure 6 below.

Forests outside the forest area have a considerable loss potential. The government does not have significant authority over forest land outside the forest area, but they have a considerable responsibility for forests within the forest area. Therefore, there is great potential for damage to the forest in the future. Periodic monitoring of the condition of existing forests is necessary within and outside the forest area.

*3.2. Identification of Illegal Logging Events*

The destruction of forests can be attributed to many factors, including illegal logging [39]. Illegal logging is one of the problems that leads to forest land loss in various countries, including Indonesia. Illegal logging has two main causes: industrial needs and the transition of forest functions. Firstly, the industrial demand for wood (i.e., building materials, firewood, paper, tissues, and several types of product packaging) is remarkably high. Secondly, some parties previously cleared the land by cutting down trees to

change forest functions (i.e., the transfer of forest functions to oil palm land, settlements, agriculture, and other functions) [40,41].

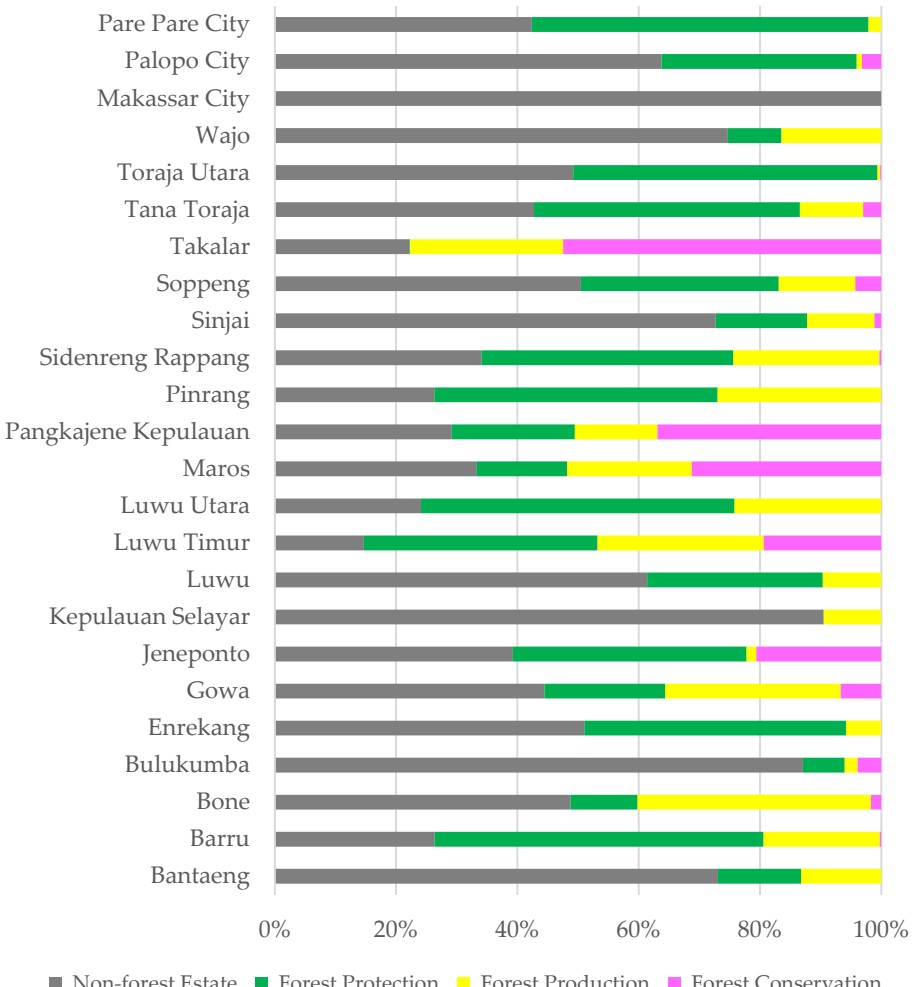

**Figure 6.** Percentage of Forest Area in Designated Forest Areas.

The information obtained revealed that the incidence of illegal logging increased during the COVID-19 pandemic due to insufficient monitoring by the responsible parties. Research related to environmental law enforcement against environmental damage explained that the following factors influenced the high level of illegal logging in Indonesia [42].

a. Legal Factors: From a legal perspective, the governing rules are considered to be inadequate and have failed to deal with the eradication of forest destruction effectively.

b. Law Enforcement Factors. These factors are still insufficient considering the number of the existing forest police. The limitations of law enforcement officers in the regions and the lack of coordination cause problems in environmental law enforcement.

c. Facilities and Infrastructure Factors. Law enforcement, including highly educated and skilled human workers, good organisation, and sufficient equipment and finances, among others, is difficult to realise regardless of support for adequate facilities.

d. Community Factors. The social and cultural angles of society in Indonesia are divided into two, namely the upper class (rich people) and the lower class (poor people). Law enforcement between the two is also substantially different because of a different mindset and knowledge. If a person is at the bottom, then their desire or obedience to a law by a person is highly unlikely or they may be unwilling to obey the laws that have been regulated due to limited knowledge and education. Thus, the person may be unaware of the binding sanctions if violated.

The four aforementioned conditions illustrate the weak enforcement of forestry and environmental laws in the current technological developments. The era of Forestry 4.0 should facilitate easy forestry monitoring, especially for data acquisition and analysis [43,44]. GEE technology in this study is one of the breakthroughs for overcoming these problems. Monitoring through digitisation can directly minimise monitoring movement in the field. Moreover, monitoring the results on GEE could be the basis for field monitoring on areas with high incidence rates. Therefore, socialisation in the community could be emphasised. The identification results of illegal logging events using the GEE platform obtained 1971 spots of forest change events in the vulnerable period of the first period (January–April) with the second period (April–August), and 1680 spots of forest change events in the second period (April–August) with the third period (September–December), with a total incidence area of 7599.28 ha, as presented in Figure 7.

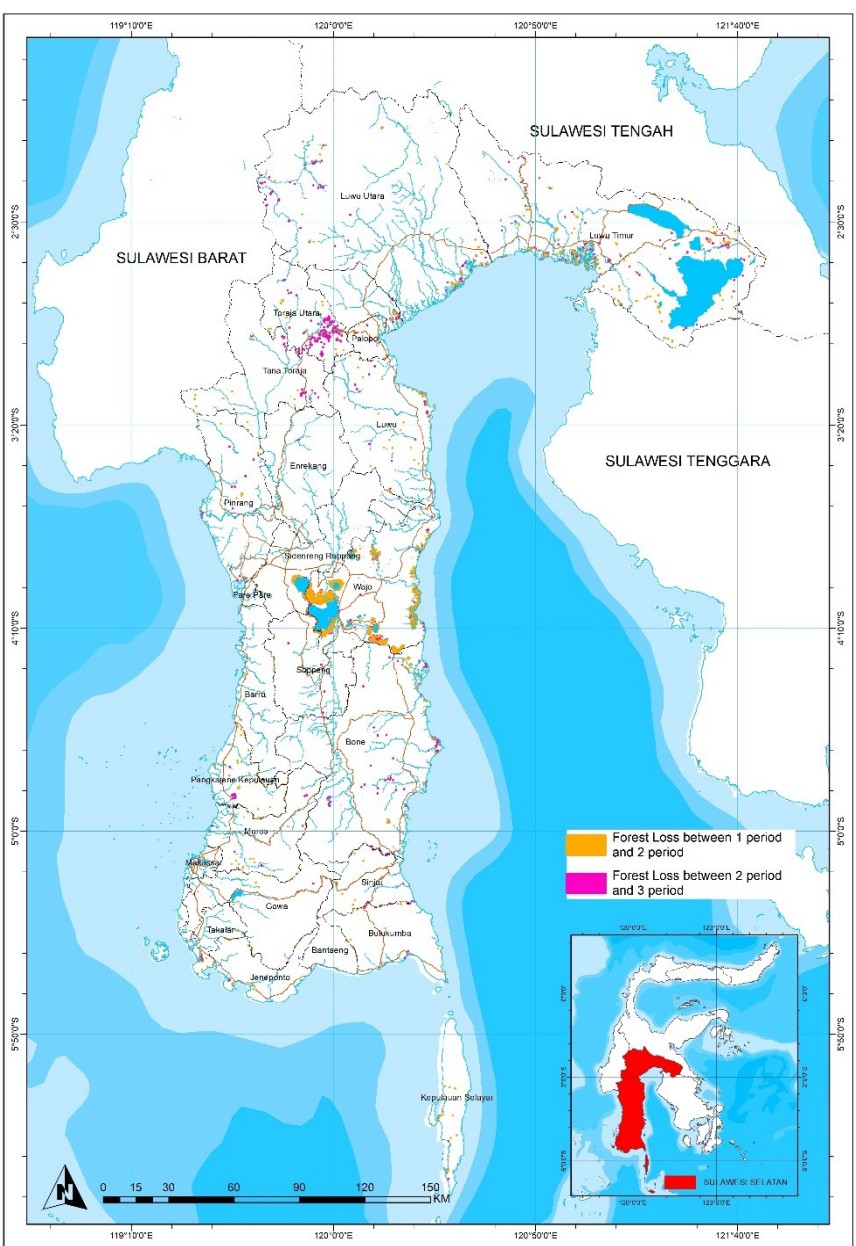

**Figure 7.** Distribution map of illegal logging events in Sulawesi Selatan in 2021.

After conducting an accuracy test of illegal logging events obtained from GEE with illegal logging events monitored by the Sulawesi Regional Environment and Forestry Law

Enforcement Centre, most of the illegal logging events monitored in the field (Figure 8) revealed a high conformity, as presented in Table 1, with the results of illegal logging identification produced from the GEE platform, that is, an accuracy kappa value of 0.89 [33,35].

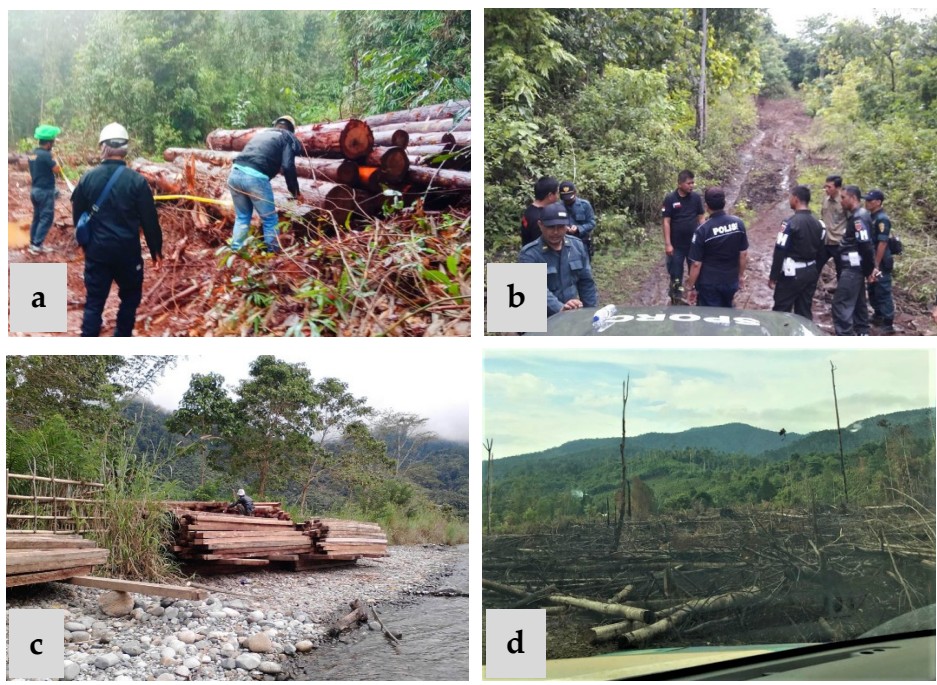

**Figure 8.** Layer catch of illegal logging incidents in Sulawesi Selatan: (**a**) authorities confiscated illegal logging timbre in East Luwu Regency, (**b**) illegal logging actors made roads for illegal timbre skidding in Takalar Regency, (**c**) confiscation of illegal logging timbre in Gowa Regency, and (**d**) illegal logging of forests to be converted into pepper plantations in East Luwu Regency.

**Table 1.** Confusion matrix for illegal logging classified map.

| | | Reference Data | | | |
|---|---|---|---|---|---|
| | Class | Illegal Logging Incident | Not Illegal Logging | Total | User's Accuracy |
| Model | Illegal Logging Incident | 360 | 11 | 371 | 97.04 |
| | Not Illegal Logging | 10 | 119 | 129 | 92.25 |
| | Total | 370 | 130 | 500 | 189 |
| | Producer Accuracy | 97 | 8 | 97 | 479 |

Overall Accuracy: 95.80

Kappa Cofficient: 0.89

The results of this illegal logging identification are in accordance with secondary data on field monitoring conducted by the relevant agencies. Several regencies have the potential to experience high illegal logging events, including Bone Regency, Jeneponto, Selayar, Luwu, Luwu Timur, Luwu Utara, Maros, Pangkajene Kepulauan, Pinrang, Sidenreng Rappang, Soppeng, Takalar, Tana Toraja, Toraja Utara, and Wajo. The location of this illegal logging incident is in line with the research of Rijal on Profile, Level of Vulnerability, and Spatial Pattern of Deforestation in Sulawesi Period of 1990 to 2018 [45], wherein several Sulawesi Selatan regencies have a level of deforestation vulnerability, including Luwu Timur, Luwu Utara, Luwu, Pinrang, Selayar, Toraja Utara, Pangkajene Kepualauan, Sidenreng Rappang, and Jeneponto. Most of the illegal logging incidents in Sulawesi Selatan occur in undesignated forest areas, as shown in Figure 9.

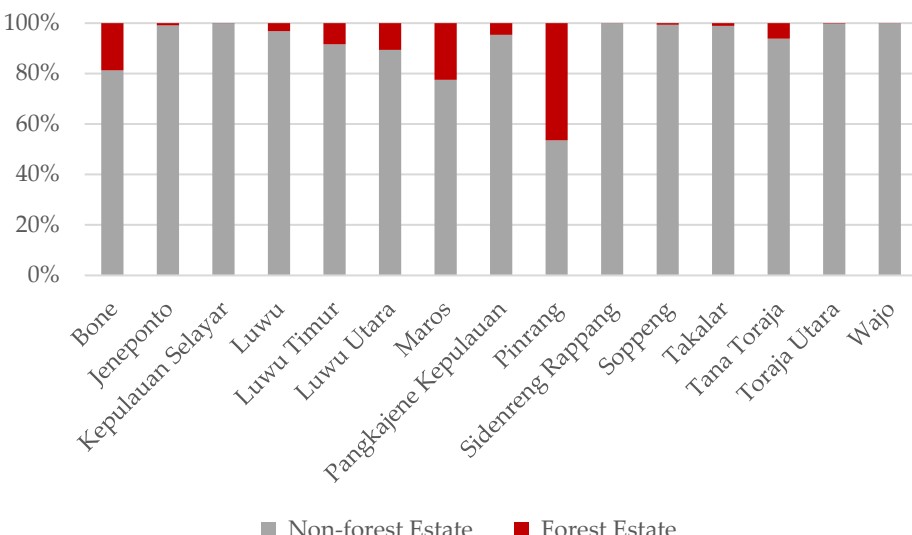

**Figure 9.** Comparison of the incidence of illegal logging within forest and non-forest areas in Sulawesi Selatan Province.

## 4. Conclusions

A formula was produced in this study to identify the occurrence of illegal logging using the GEE platform. The occurring problems of environmental and forestry monitoring, such as the lack of monitoring due to limited personnel, facilities, and infrastructure, can be resolved with this technology. The accuracy level of identification using Sentinel 1 SAR and Sentinel 2 multispectral images provided the best results (kappa = 0.89). The identification results of illegal logging events using the GEE platform obtained 1971 spots of forest change events in the vulnerable period of the first period (January–April), with the second period (April–August), and 1680 spots of forest change events in the second period (April–August) with the third period (September–December) with a total incidence area of 7599.28 ha. The use of GEE and Sentinel imagery to monitor forest conditions in Indonesia is highly recommended to minimise the shortage. Monitoring personnel can use the results of this analysis to prioritise locations that must be monitored (for example, by monitoring locations where illegal logging events are high based on monitoring remote sensing technology in the form of satellite imagery).

**Author Contributions:** Conceptualisation, A.M. and M.N.; methodology, A.M. and M.N.; software, M.N.; validation, A.M., M.N. and A.S.S.; formal analysis, M.N.; investigation, M.N.; resources, M.N.; data curation, M.N.; writing—original draught preparation, A.M. and M.N.; writing—review and editing, A.M., M.N. and A.S.S.; visualisation, M.N.; supervision, A.M. and A.S.S.; project administration, A.M.; funding acquisition, A.M. All authors have read and agreed to the published version of the manuscript.

**Funding:** This research received no external funding.

**Data Availability Statement:** Not applicable.

**Acknowledgments:** The authors would like to express their deepest gratitude to all parties involved for their help. The author also expresses his gratitude to the Sulawesi Region Environmental and Forestry Law Enforcement Centre for their support.

**Conflicts of Interest:** The authors declare no conflict of interest.

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
