# Peer review of "Monitoring Illegal Logging Using Google Earth Engine in Sulawesi Selatan Tropical Forest, Indonesia"

_forests, doi:10.3390/f14030652_

Round 1

Reviewer 1 Report

This paper has an interesting topic, but it is still not feasible to proceed to the next process. Several researchers have published the use of GEE in detecting deforestation, including the effects of illegal logging. In the introduction, the authors do not justify the research's importance. There is no visible improvement in the use of GEE in similar cases. In this section, the authors should convey the advantages of using Sentinel images in detecting forest-cover changes over a certain period compared to identical satellite images. Information on Sentinel's resolution and repeat cycle will be important in the speed of detecting changes in forest cover.

Weaknesses are most often found in the method section. In the method section, the authors should include a flow chart of the method used in this study (Line 117). Regarding the scope of the study, the authors should limit the research location only to the state forest area (Line 86). The definition of illegal logging is not stated in the method, and illegal logging should be limited only to state forest areas that previously had forest cover. The authors do not explain the number of sentinel scenes used and the date of coverage in each observation period (3 periods) (Line 96). The period of observing images is too long (4 months for each period). Illegal logging detection should be carried out between the shortest possible periods, for example, every two weeks or every month following the cycle/duration of satellite imagery. This aims to quickly detect illegal logging incidents so that it helps overcome the lack of human resources in the field.

The validation method is also not appropriately explained in how authors validate: a. Forest cover Vs non-forest cover; b. deforestation due to Illegal logging Vs deforestation due to non-illegal logging. How many of each is the sample number, and how is it determined?

This paper is also weak in terms of citations. Many statements are not supported by related references, for example, in lines 67-72, 89-93, 95, 128-141, 153-156

In the results section, there are 1971 and 1680 spots of illegal logging with a total area of 7599.28 ha. There is no explanation for the spot number and the area of illegal logging events monitored by the Sulawesi Regional Environment and Forestry Law Enforcement Centre. The authors only show a few points of illegal logging incidents on the map of research locations (Figure 1). Based on the map of the illegal logging area, this paper will be more useful if the authors can provide forest conditions that are prone to illegal logging incidents.

Some paragraphs are not related to the topic of discussion, such as in lines 215-248 (should convey the results of forest cover identification in 3 periods along with validation), lines 258-281 (should display illegal logging areas between the two periods, accompanied by validation results), line 312- 316.

Based on the weaknesses of the method used and deficiencies in the presentation of the analysis, the authors had to re-analyze and be able to rearrange the manuscript.

Author Response

  1. Comment 1 regarding the importance of research we add in the introduction, that for tropical regions such as Indonesia the use of Sentinel imagery data for forest monitoring is very new and is indeed very much needed to make observations in an area without any cloud disturbance.
  2. Comment 2, this research is indeed limited to forested areas in the forest area and we focus on forest areas in South Sulawesi Province, as we present at the research location. In GEE, we immediately issued data information per study area so that the image scene was not depicted. This is indeed different if we download directly to the image provider's website. For the duration of monitoring, we do it to see the conditions in advance in a year so we divide it into several observation periods to validate it with illegal logging monitoring data in the field. But it's a good suggestion to do it every month, and we will develop the algorithm on GEE even further in future research.
  3. Comment 3 regarding the validation performed in the GEE platform using the Kappa accuracy test. As for the accuracy test data, it uses data from related agencies in Indonesia that monitor illegal logging incidents in the field. We then analyze the event monitoring point data to measure the accuracy of the model.
  4. Comment 5, regarding illegal logging monitoring points obtained from the Sulawesi Regional Environment and Forestry Law Enforcement Centre, the number is presented in the validation section.
  5. Comment 6, we think the explanation regarding the results in the Identification of Illegal Logging Events section is a reflection of the results of the research described in the two observation periods.

Reviewer 2 Report

The authors used sentinel 1 and 2 for monitoring logging in within the google earth engine (GEE). As for me, the topic is so interesting but the novelties are not clear and the presentation could be much better it is. I have some major revisions that the author should be addressed in the paper up to this could be more intersting and publishable. 

1- The abstract was not well presented, especially its first 5 lines. Please highlights key content areas, your research purpose, the relevance or importance of your work, and the main outcomes. 

2- The novelties of the presented manuscript are limited or not clearly mentioned in the last paragraph of the introduction. please clarify the novelties in more detail in the last paragraph of the paper. the early monitoring of illegal logging is one of the most important case studies in GEE, as well. please see the following link: 

https://earthengine.google.com/case_studies/

 3- please cite the following paper for the explanation on Page 2, Line --. 

https://doi.org/10.1109/JSTARS.2020.3021052

https://doi.org/10.1016/j.isprsjprs.2020.04.001

4- I couldn't see any workflow for the methodology. The methodology is the heart of the paper that didn't well present. 

5- The author used some GEE script in the paper that is not commented on in the scientific journal. please refer readers to the available code on the specific URL. 

6- About the validation, Please shows the condition matrix in a table. 

Author Response

  1. Comment 1, we provide an explanation regarding the importance of this research that the lack of monitoring of overall forest conditions is the impact of the current high forest destruction. Through this research, the problems encountered related to environmental damage due to illegal logging will be illustrated through the use of remote sensing technology which is currently highly developed based on artificial intelligence and machine learning, namely the Google Earth Engine (GEE).
  2. Comment 2 regarding the novelty of this research, is that this is a new-research in the use of cloud computing technology, especially in Indonesia for forest monitoring. Of course, in the future this platform will not only serve as research but will be used by related agencies for their work.
  3. Comment 3, regarding the suggested article, we have added it as an explanation in our manuscript.
  4. Comment 6, validation is carried out in the GEE platform which immediately issues the accuracy test value.

Reviewer 3 Report

Overall, the article Monitoring Illegal Logging using Google Earth Engine in Sulawesi Selatan Tropical Forest, Indonesia can be accepted with correction as mentioned below.

1. Authors are required to add one more section that discussed about the Related work or Literature Review. This section should be written as 2.0 Related Work. the 3.0 Materials and Methods and follow to other section.

2. Materials and Methods, is good if author can draw a methodology diagram and explain it. Easy for another researcher to understand. 

3. Figure 5, the colour used to differentiate between Non-Forest Estate and Forest Protection need to be change. 

4. Need to clarify whether the statement "The era of forestry 4.0 should facilitate easy forestry monitoring, especially in data acquisition and analysis [42, 43]." Please confirm "the era of forestry 4.0 ..."

5. "..... the average comparison of VV and VH polarisation images, ...." Please state what is VV and VH.

6. Is good if author can show the results in Table form and explain from that Table. 

7. "Most of the illegal logging incidents in Sulawesi Selatan occur in undesignated forest areas, as shown in Figure 8." There is no Figure 8 in this article. Please read back and author required to improve it. 

Author Response

  1. Comments on section 3, for the colors between non-forest estate and forest protection are different.
  2. Comments on section 4, Forestry Era 4.0 is a forestry development concept that is being promoted in Indonesia as support for the industrial era 4.0. Where this era is expected in management and forestry to keep up with the times by using existing technological developments, especially in the field of mapping which is developing based on cloud computing.
  3. Comments on section 5, VV and VH are the polarization of Sentinel 1 radar images. VV where the satellite transmits waves vertically and receives vertically while VH where the satellite transmits waves vertically and receives horizontally. This polarization is a band to extract object information. However, because the Radar Image only has one channel, to extract the information is to adjust the polarization.
  4. Comments on section 7, for Figure 8 have been corrected

Reviewer 4 Report

In the manuscript entitled "Monitoring Illegal Logging using Google Earth Engine in Sulawesi Selatan Tropical Forest, Indonesia" submitted to Forest journal, the authors used Google Earth Engine (GEE) to monitor illegal logging. The authors used the random forest classification algorithm of the GEE platform to obtain data on forest conditions for 2021, covering an area of 2,843,938.87 ha or 63% of the total area of Sulawesi Selatan Province. The article is within the Forest scope and provides useful information for the journal readers. I recommend the publication after major revisions.

1.    Abstract section is not summarizing well the key points of the research work. The findings/importance of this research should be presented in one sentence at the end of this abstract section. Please clarify the contradictory meaning of the following statements: Getting satellite imagery with relatively small cloud cover for tropical regions, such as Indonesia, is remarkably difficult. This difficulty is due to the presence of a radar sensor on Sentinel 1 images that can penetrate clouds, allowing the observation of the forest condition even in the presence of clouds.

2.    The authors reviewed the state of the art regarding the works done in the past and the current needs of improvement, in the introduction section. Here are some other articles to improve this section: https://www.frontiersin.org/articles/10.3389/ffgc.2022.1018762/full, https://www.sciencedirect.com/science/article/pii/S0303243421002397, https://www.mdpi.com/1999-4907/11/12/1283, https://www.fs.usda.gov/rm/pubs_journals/2021/rmrs_2021_chen_s001.pdf.

3.    The data collection section should be rewritten, better explained. The GEE platform is adequate to the purpose of the article. The datasets are covering the Sulawesi Selatan Province of Indonesia covering an area of 2,843,938.87 ha of tropical forest.

4.    In the results and discussion section, the authors successfully determined the forest cover and the illegal logging sites. Their results show that most of the illegal logging events monitored in the field revealed a high conformity with the results of illegal logging identification produced from their GEE platform.

5.    The key research conclusions should be more emphasized. Conclusion section should be extended and the correlation to the research results better explained.

7.       The references cited must be checked again for the journal standards. Reference 5 is not clear.

Author Response

  1. Comments to part 1, Indonesia is a tropical country where land monitoring using remote sensing data has many obstacles, especially in multispectral images where almost every time every area, especially forested areas, is always covered with clouds. So that the presence of radar images such as Sentinel 1 can help to overcome these problems because Sentinel 1 images can minimize cloud disturbances. With the availability of this image database on GEE it is very helpful in monitoring. This research aims to optimize the role of GIS platforms such as GEE to be used in monitoring the condition of areas, especially forests.
  2. Comments 2, 5 and 7, we make improvements to the citations and add explanations.

Round 2

Reviewer 1 Report

Authors did not make improvements as suggested, especially in the method section.

Author Response

Thank you for your comments. We have made improvements to the manuscript that we have revised.

Reviewer 2 Report

Please add a flowchart or workflow for the presented work and explain it in more detail. Moreover, Please address my last comments in more detail. Please add the accuracy table for each classes like user and produce. please show the training and test data. do you have test data. 

Author Response

Thank you for your comments. We have made improvements to the manuscript that we have revised. We have added a research framework and an accuracy matrix of validation activities.

Reviewer 4 Report

1.    The data collection section should be rewritten, better explained. Not even a word was changed in 2.1. Data Collection section

5.    The key research conclusions should be more emphasized. Conclusion section should be extended and the correlation to the research results better explained. Not even a word was changed.

7.       Reference 5 is not clear. You made no corrections.

Author Response

Thank you for your comments. We have made improvements to the manuscript that we have revised. We have revised in sections 1, 5 and 7.